# Immunity as Cornerstone of Non-Alcoholic Fatty Liver Disease: The Contribution of Oxidative Stress in the Disease Progression

**DOI:** 10.3390/ijms22010436

**Published:** 2021-01-04

**Authors:** Marcello Dallio, Moris Sangineto, Mario Romeo, Rosanna Villani, Antonino Davide Romano, Carmelina Loguercio, Gaetano Serviddio, Alessandro Federico

**Affiliations:** 1Hepatogastroenterology Division, Department of Precision Medicine, University of Campania “Luigi Vanvitelli”, Via S. Pansini 5, 80131 Naples, Italy; marcello.dallio@unicampania.it (M.D.); mario.romeo@unicampania.it (M.R.); carmelina.loguercio@unicampania.it (C.L.); alessandro.federico@unicampania.it (A.F.); 2C.U.R.E. (University Center for Liver Disease Research and Treatment), Liver Unit, Department of Medical and Surgical Sciences, University of Foggia, 71121 Foggia, Italy; rosanna.villani@unifg.it (R.V.); antoninodavide.romano@unifg.it (A.D.R.); gaetano.serviddio@unifg.it (G.S.)

**Keywords:** non-alcoholic fatty liver disease, trained immunity, oxidative stress, hepatocellular carcinoma

## Abstract

Non-alcoholic fatty liver disease (NAFLD) is considered the hepatic manifestation of metabolic syndrome and has become the major cause of chronic liver disease, especially in western countries. NAFLD encompasses a wide spectrum of hepatic histological alterations, from simple steatosis to steatohepatitis and cirrhosis with a potential development of hepatocellular carcinoma. Non-alcoholic steatohepatitis (NASH) is characterized by lobular inflammation and fibrosis. Several studies reported that insulin resistance, redox unbalance, inflammation, and lipid metabolism dysregulation are involved in NAFLD progression. However, the mechanisms beyond the evolution of simple steatosis to NASH are not clearly understood yet. Recent findings suggest that different oxidized products, such as lipids, cholesterol, aldehydes and other macromolecules could drive the inflammation onset. On the other hand, new evidence indicates innate and adaptive immunity activation as the driving force in establishing liver inflammation and fibrosis. In this review, we discuss how immunity, triggered by oxidative products and promoting in turn oxidative stress in a vicious cycle, fuels NAFLD progression. Furthermore, we explored the emerging importance of immune cell metabolism in determining inflammation, describing the potential application of trained immune discoveries in the NASH pathological context.

## 1. Introduction

Non-alcoholic fatty liver disease (NAFLD) is defined as the pathological hepatocyte fat accumulation in >5% of liver tissue, in absence of viral hepatopathy, alcohol consumption, drug intake, and other secondary causes [1]. A wide spectrum of conditions exists, ranging from the “simple” hepatocyte fat accumulation to a more complex histologic picture characterized by cytolytic damage, lobular inflammation and fibrosis [2], namely non-alcoholic steatohepatitis (NASH). It can be complicated by the development of cirrhosis and/or hepatocellular carcinoma (HCC), determining hugely the worsening of the prognosis [1,2]. Therefore, due to the high incidence in western countries and the intricate clinical management, NAFLD has become one of the most important health care burden of the 21st century [1,2].

NAFLD has been studied as the hepatic complication of the metabolic syndrome (MS) as well as an independent risk factor for the development of cardiovascular diseases [1,3,4]. In fact, the major risk factors for NAFLD development and progression represented the pathologic background that defines MS, determining a further increase of the cardiovascular risk in a vicious cycle [4,5,6]. Therefore, in a scenario in which the prevalence of MS increases [7] and viral hepatitis are efficiently contrasted, NAFLD has become the first cause of liver transplantation in the western countries [1,7,8]. Nowadays it constitutes the global predominant hepatopathy showing a prevalence of 6–35% worldwide (25–26% in Europe), with wide variations due to the ethnicity [8,9].

From a pathogenetic standpoint, NAFLD is considered the result of a multifactorial interaction among genetic background (including genomic instability, single nucleotide polymorphisms (SNPs), microRNA expression), environmental factors and dysmetabolism, with a prominent role played by insulin resistance (IR) [10,11] (Figure 1).

Accordingly, a recent expert consensus suggested that metabolic (dysfunction) associated fatty liver disease (MAFLD) would be a more appropriate definition [12]. The main factor in NAFLD pathogenesis has been identified in the hepatic accumulation of free fatty acids (FFAs) [13]. Mitochondria activity counteracts this FFAs overload, increasing the inner membrane permeability with the loss of its biologic function [14]. The direct consequence of this impairment is the mitochondrial dysfunction associated with the reactive species production, responsible in turn, of several consequences such as mtDNA mutations, lipotoxic intermediates accumulation, pro-inflammatory cytokines production, which promote IR, immune response and hence disease progression [15]. Moreover, new evidence indicates innate and adaptive immunity activation as the driving forces in establishing liver inflammation and fibrosis [16,17] (Figure 1).

In this review, we discuss the mechanisms underlying the crosstalk between oxidative stress and immune dysfunction in NAFLD evolution with new insights in the concepts of immune cell metabolism and trained immunity.

### 1.1. Overview on Molecular Mechanisms of Oxidative Stress in NAFLD

Chronic liver diseases are characterized by excessive production of reactive species and redox imbalance [18]. NAFLD is a very complex disease that involves several factors, genetic, epigenetic and environmental. Currently, the “multiple hit” pathogenesis is the accepted theory to explain its development, giving a mechanistic overview in which these factors act in parallel [19]. In this model, oxidative stress works as a cornerstone in NAFLD progression. Normally reactive species mediate a plethora of cellular functions such as apoptosis, proliferation, metabolism and immune defense regulation [20,21]. Conversely, the excessive reactive oxygen species (ROS) production leads to oxidative stress, compromising the physiological redox signaling with the consequent cellular damage [22,23]. This phenomenon acquires even more relevance in case of powerful ROS production, generated in a tissue oxidative microenvironment [24]. A typical example is represented by the iron overload which occurs in some NASH patients, facilitating the Fenton reaction that generates the extremely toxic HO from H_2_O_2_ [14].

Liver possesses potent scavenging systems to counteract the production of ROS deriving especially from mitochondria and cytochrome P450 (CYP) as well as other enzymes. However, when the ROS production overcomes the scavenging capacity, a redox imbalance occurs and superoxide anion radicals (O_2_^·−^) and hydrogen peroxide (H_2_O_2_) are continuously formed as bioproducts of energetic metabolism [14,25]. This involves hepatocytes, Kupffer cells (KCs), neutrophils and other liver cell types, underlying the crosstalk between oxidative stress and immunity in the progression of the disease. Moreover, FFAs deriving from diet, released by the adipose tissue or produced by de novo lipogenesis, overload the liver, serving as substrate for production of lipotoxic substances (e.g., ceramides, lysophosphatidylcholines and diacylglycerols) especially by mitochondria and CYP activities [26,27,28,29,30]. This metabolic stress promotes pro-inflammatory cytokine release and cell death that, in turn, represents one of the most important mechanisms of damage in the NAFLD histologic picture [31,32]. ROS can be generated in several organelles, such as mitochondria, endoplasmic reticulum (ER) and peroxisomes, but also by a lot of enzymes such as xanthine oxidase, CYP2E1, cyclooxygenases, NADPH oxidase (NOX) and lipoxygenase [20,33].

#### 1.1.1. Oxidative Stress Derived from Mitochondrial Dysfunction

The most important source of ROS in NAFLD is represented by mitochondria, that also act by responsible of lipid metabolism dysregulation in case of impaired biologic functioning typically highlighted in metabolic disorders [34,35]. Accordingly, alterations in oxygen consumption, electron transport chain (ETC) activity and fatty acid oxidation (FAO) are widely described in literature. In fact, during early stages of NAFLD, the liver faces a high influx of external FAs and several studies showed the increase of FAO in different animal models of NASH [36,37,38,39,40], although some contrasting data exist in human ones [41,42,43,44,45,46,47,48,49,50,51,52]. A recent study, for instance, described the raising of FAO after 4 weeks in high fat diet (HFD)-fed mice, while after 8 weeks the normal β-oxidation was restored [53].

In the lipid-rich condition indeed, the liver tries to counteract the excessive production of lipotoxic molecules, by accumulating triglycerides and controlling FAO [24,54,55], even if several factors such as IR, leptin and enterokines might also play a role in this scenario [56]. ETC results also altered by FAs overload as the presence of FAs, especially Polyunsaturated fatty acids (PUFAs), generates O_2_^·−^ and inactivates complex I and III [57,58,59,60], by direct interaction with ETC components or interference with membrane fluidity which provokes proton leakage [57,58,59,60,61,62]. The role of oxysterols is NAFLD progression should not be underestimated as suggested by novel evidence [63]. In fact, in a recent study it was shown that cholesterol supplementation in HFD-fed mice determined the hepatic accumulation of specific oxysterols (e.g., 7-hydroxycholesterol, 7-Ketocholesterol, and 5-cholestane-3, 5, 6-triol), which acted synergistically with FAs to impair mitochondrial function and biogenesis, facilitating NAFLD progression [27].

Several further discrepancies exist in literature about the mitochondrial dysfunction in simple steatosis and NASH such as alterations of cytochrome c oxidase (COX) activity [41,64,65,66,67,68,69], oxygen consumption of isolated mitochondria [70,71,72], oxidative phosphorylation (OXPHOS) efficiency [68,71,72] and ATP levels [72,73,74,75,76], either in patients and murine models. In particular, COX activity resulted unchanged, reduced or increased; ATP levels have been described as normal or reduced; mitochondrial oxygen uptake was augmented, normal or largely decreased; OXPHOS resulted augmented or unchanged [64,65,66,67,68,69,70,72,73,74,75,76].

#### 1.1.2. ER Contribution to Oxidative Stress

An important contribute to oxidative stress derives also from ER stress. For instance, the production of H_2_O_2_ derives from transfer of electrons to molecular oxygen operated by ER oxidoreductase-1, which in turn received electrons from protein disulphide isomerase in the process of protein folding [24]. Several animal studies showed that the increased activity of C/EBP homologous protein (CHOP) is involved in the UPR-mediated ROS production during prolonged ER stress [77,78]. Disturbances in GSH/GSSH ratio in ER lumen entails protein misfolding and ROS production [79,80,81]. Furthermore, RNA-dependent protein kinase-like ER eukaryotic initiation factor-2 α kinase (eIF2a) phosphorylates nuclear factor erythroid 2–related factor *2* (Nrf2), promoting its translocation to the nucleus and the expression of antioxidants weapons such as hemeoxygenase-1 (HO-1) [82]. Therefore, the crosstalk between ER stress and mitochondrial dysfunction is of current interest and needs further studies to assess its specific contribution in this context.

## 2. Redox Biology and Immune Regulation in NAFLD Progression

Redox balance is primary involved in both adaptive and innate immunity, contributing to macrophage, lymphocyte and dendritic cell (DC) signaling and in cytokine response modulation [83]. The first evidence of oxidant role in immune activity was described in 1991, when it was shown that H_2_O_2_ at micromolar concentrations induced nuclear factor kappa-light-chain-enhancer of activated B cells (NF-κB) transcription in human T lymphocytes [84].

In several liver diseases the crosstalk between redox imbalance and immune response has been described [85]. Inflammation and oxidative stress occur simultaneously in the liver, fueling each other. Moreover, persistent oxidative stress promotes the establishment of a chronic inflammation that represents the fil rouge for the detrimental interconnection between the liver disease and its systemic complications.

In fact, reactive species activate several transcriptional factors and receptors such as NF-κB, activator protein-1 (AP-1), p53, hypoxia-inducible factor 1-alpha (HIF-1α), peroxisome proliferator-activated receptor γ(PPAR-γ), β-catenin/Wnt, and Nrf2, which in turn modulate the expression of molecules involved in inflammation [86]. Of note, NF-κB is a key-mediator of ROS proinflammatory effect, being activated directly by reactive species and indirectly by ROS-damaged DNA [87].

In NAFLD, the excessive lipid accumulation is the primum movens of metabolic stress, which promotes innate immune activation, the driving force of inflammation in NASH [88,89,90]. In particular, KCs are activated by ROS and other stimuli and become the main hepatic producers of ROS, as well as cytokines, growth factors and chemokines [91,92,93].

### 2.1. The Role of PRRs

In NAFLD, the liver is continuously invaded by pathogen associated molecular patterns (PAMPs) (e.g., bacterial products, lipopolysaccharides—LPS), damage-associated molecular patterns (DAMPs) and metabolites (e.g., FFAs, mitochondrial DNA, lipotoxic products) which are sensed by pattern recognition receptors (PRRs), such as toll-like receptors (TLRs) and nucleotide-binding oligomerization domain-like receptors (NLRs) with the consequent activation of pro-inflammatory cascades [94,95,96,97]. The most known DAMP released by damaged cells is high mobility group box 1 protein (HMGB1), which activates TLR4 with the consequent myd-88 dependent NF-kB pathway induction [98,99]. In a murine model of ischemia/reperfusion damage, superoxide anion induced NOX activation by TLR4 in neutrophils [100]. Several experimental murine models showed that interaction between mtDNA and TLR-9 induced cytokine and fibrogenic responses in KCs and hepatic stellate cells (HSCs) [101,102,103]. However, a group of cytoplasmic PRRs is constituted by oligoadenylate synthase (OAS)-like receptors (OLRs), sensors of nucleic acids. In case of damage, dsDNA, microbial and mtDNA can be detected by OLRs, which activate transcription of proinflammatory genes by inhibitor of nuclear factor kappa-B kinase subunit ε (IKKε) and activator protein 1 (TBK1) signaling [17].

In hepatic inflamed areas and on HSCs there is a higher expression of receptors for advanced glycation end-products (RAGEs), a class of PRRs [104]. The AGEs formation promotes the generation of a considerable quantity of ROS; furthermore, oxidized RAGEs activate NOX1, contributing to ROS production [105]. NLR family pyrin domain containing 3 (NLRP3) is a PRR that interacts with PAMPs and DAMPs and is involved in generation of inflammasome complexes. In particular, ROS produced by mitochondria and NOXs promote NLRP3 inflammasome formation [106], while scavengers such as catalase and superoxide dismutase (SOD) inhibit its activation [107]. Accordingly, it has been suggested that the crystal structure of NLRP3 is very sensitive to red ox balance variations [108].

Overall, PRRs activate several intracellular targets such as Transforming Growth Factor-β (TGF-β) activated kinase 1 (TAK1) and apoptosis signal-regulating kinase 1 (ASK1), both members of mitogen-activated protein kinase kinase kinases (MAP3K) family, which activate in turn other kinases such as c-Jun-N-terminal chinasi (JNK), adenosine monophosphate-activated protein kinase (AMPK) and IkB, hence with a key role in NASH pathogenesis [109,110,111].

Interestingly, apart from receptor interaction, several reports described that ROS could influence downstream innate immunity signaling [112]. For instance, H_2_O_2_ increases activity of MAPKs (e.g., p38) and therefore, JNK pathway, which in turn promotes the production of further ROS [113,114,115]. Moreover, NF-kB, AP-1 and PPARs can be directly modulated by oxidative species, inducing transcription of cytokines [116,117,118,119]. ROS and lipid peroxidation products such as hydroxinonenal- (HNE-) and malondialdehyde- (MDA-) adducts can activate NF-kB directly or by inhibition of IKKs phosphorylation in innate immune cells [120,121,122]. Therefore, the natural consequence of persistent stimuli is the chronic inflammation, activation of HSCs with collagen deposition, increased IR and further aberrant expression of pro- and antioxidant enzymes [123,124,125]. However, the presence of redox imbalance and lipid peroxidation is a common characteristic of NAFLD and NASH [26] and the involvement of adaptive immunity seems to be a key factor in NAFLD progression. Accordingly, recently Azzimato V. et al. described that liver macrophages are proinflammatory in NASH, but not in NAFLD and obesity, although sings of oxidative stress are present in both conditions as NRF2 is downregulated [126].

### 2.2. The Role of OSEs

Oxidative stress has been described as the main trigger in stimulating adaptive immunity, especially through oxidized phospholipids and aldehydes generated by lipid peroxidation such as MDA, malondialdehyde-acetaldehyde (MAA) and 4-HNE-protein adducts, formally named oxidation-specific epitopes (OSEs) [127,128,129]. OSEs are implicated in several systemic diseases and atherosclerosis, by stimulating both innate and adaptive immunity [127,130].

Specific anti-OSE immunoglobulin G (IgG) are highly prevalent (about 40%) in adult with NAFLD or NASH and in children with NASH (60%). Moreover, high they have been associated with fibrosis, severity of hepatic lobular inflammation and intrahepatic infiltrate of B cell and T cell aggregates [128,131,132]. These observations were corroborated in animal models of NAFLD and NASH. In particular, titers of anti-OSE IgG correlated with B cell maturation in plasma cells, while antioxidant treatment with N-acetylcysteine prevented antibody response [131,133,134]. Mice pre-immunized with protein MAA-adducts present a Th1 polarization of CD4+ T lymphocytes in the liver, which produce IFN-γ, promoting KCs M1 activation [131]. Accordingly, in patients with NASH anti-OSEs IgG correlate with circulating Interferon γ (IFN-γ) levels [133]. In fact, the involvement of Th1 and Th17 lymphocytes in NAFLD evolution to NASH has been widely reported [131,135,136,137,138]. In parallel, in NAFLD murine models the quantity of hepatic CD4+ regulatory T cells (Tregs) lymphocytes is reduced [139,140]. Treg cells are typically involved in maintaining self-tolerance and controlling excessive T-cell activation in the liver [141,142]. It seems that ROS promote Treg cells apoptosis, hence facilitating Th1 and Th17 proinflammatory activity [139,140]. Therefore, the B cell activity with anti-OSE IgG production mirrors the presence of oxidative stress and inflammation. By contrast, it has been described that anti-OSEs IgM decrease in NAFLD patients compared to controls and inversely correlate with obesity and liver damage [143]. Accordingly, T15 natural IgM against oxidized phosphatidylcholine protect against NASH in LDLR KO mice fed with HF-HC diet, probably by preventing the oxidized-LDL-induced activation of KCs [144].

Overall, these findings underline how oxidative imbalance behaves as the cornerstone of NAFLD progression, fueling innate immunity to produce cytokines and new reactive species which perpetuate inflammation thorough adaptive immune cells (summary in Table 1).

## 3. The Immune Response in HCC Development

Nowadays HCC represents the fifth most common tumor and the third leading cause of cancer-related death worldwide, resulting in over 600,000 deaths annually [145]. It can occur in the NAFLD context independently to the surrounding advanced fibrosis or cirrhosis, complicating the course of the disease and worsening hugely the prognosis [146].

HCC pathogenesis still remains extremely complex, embracing various and not completely clarified biological dynamics. In the last few decades, in parallel to the genetic and epigenetic mechanisms surrounding HCC development and worsening, several findings revealed three linked elements creating a tumoral microenvironment: the chronic phlogosis, the oxidative stress and the immune system dysregulation [146,147,148].

Normally, inflammation represents the adaptive response to tissue damage, involving different mechanisms (blood vessel dilatation, immune cell recruitment, and cytokines production) and aiming to damage resolution and homeostasis recovering [149].

We have previously described the importance of the bidirectional relationship between oxidative stress and immunity in NAFLD progression. Moreover, when the inflammatory stimuli persist or the regulatory mechanisms are disrupted, a “non-resolving inflammation” occurs, determining pathological consequences such as fibrosis and metaplasia, as described in several human diseases, including HCC [149,150]. Therefore, chronic inflammation is critical for HCC development so that this tumor represents a classic paradigm of inflammation-linked cancer [151]. In line with the current scientific knowledge, several studies revealed that more than 90% of HCCs arise in the context of hepatic inflammation and, interestingly, the use of non-steroidal anti-inflammatory agents decreases its incidence and/or recurrence [149,152,153,154].

The innate immune system is triggered by oxidative stress and persistent injury to react not only against “non-self” potential pathogens but also to DAMPS, encouraging thus the emerging concept of “sterile inflammation” that seems to be very crucial in HCC development [155,156].

As previously mentioned, most of DAMPs are produced under condition of oxidative stress and/or mitochondrial dysfunction [156,157]. The high ROS production induces mtDNA damage, triggering mitochondrial dysfunction and generating thus a vicious cycle in which a deficient production of enzymes involved in the respiratory electron transport chain occurs, worsening thus ROS production [158]. After being released into the cytosol and extracellular environment, mtDNA acts as DAMP, since it contains the unmethylated CpG DNA motifs [159,160].

Furthermore, mitochondrial dysfunction is also associated to dysregulation of mitochondrial fission, mitochondrial permeability transition (MPT) pores, and necroptosis (a form of programmed cellular death), all involved in NASH and liver cancer progression [158,161,162,163].

As already described, an important DAMP is HMGB1, a ubiquitous nuclear non-histone protein that participates in DNA replication, transcription, and repair [164]. The biologic activity of released HMGB1 depends on its redox state (fully oxidized or disulphide), which is modulated by the oxidative features of the tumoral microenvironment [165].

The fully oxidized HMGB1 form promotes immunotolerance, reducing the maturation and the activation of DCs and favoring thus tumor immune escape mechanisms [165].

DCs are an important hepatic cellular population, whose alterations, in terms of reduced functionality such as reduced maturation and failure of HCC-associated antigens presentation, are involved in tumor pathogenesis [166,167,168]. In HCC patients, the amount of activated DCs (CD83-positive DCs) in hepatic nodules is lower than in cirrhotic or healthy control tissues [169]. Moreover, the lower DCs expression of human leukocyte antigen (HLA) class-Ⅰ molecules implies the loss of antigen presentation activity, thus contributing to impaired recruitment of tumor-specific lymphocytes [170]. Considering the major function of DCs in the interplay between innate and adaptive immune reactions, the disruption of the efficiency of these mechanisms causes a weak T cell immune response against cancer [171]. Accordingly, CD14+ cytotoxic T-lymphocyte-associated protein (CTLA)-4+ DCs, a subset recently identified, are over-represented in tumoral tissue of HCC patients and are involved in the production of some regulatory cytokines (e.g., IL-10), which suppress the CD4+ T-cell immune response, facilitating the tumor progression [172].

As already discussed, HMGB1 disulphide form behaves as a DAMP, activating TLR4, and promoting innate immune response with pro-inflammatory and chemoattractant effects [173].

The activation of TLRs, through the stimulation of JNK and NF-kB pathways, leads to different consequences, including among others, the production of Tumor Necrosis Factor-α (TNF-α) (able to induce the proliferation of tumor cells) as well as many other mediators involved in the recruitment and polarization of different inflammatory cell types [174,175,176,177]. In particular, high levels of macrophage colony-stimulating factor (MCSF) and chemokine (C-C motif) ligand 2 (CCL2), vascular endothelial growth factor (VEGF), and TGF-β induce macrophages recruitment and infiltration in peri-tumoral tissue, their polarization in M2 phenotype as well as the differentiation in tumor-associated macrophages (TAMs) [178,179] (Figure 2).

Inflammation enchained with oxidative stress (both constantly present in the NASH context), mitochondrial dysfunction (including mtDNA releasing), and gut microbiota alterations (particularly, the action of LPS) all promote the creation of DAMPS and PAMPS. These molecules, in turn, are able to stimulate and activate TLRs and Inflammasomes resulting in cytokines, chemokines, and growth factors production. These events implicate several consequences and in particular: (i) monocytes recruitment, activation, and differentiation in TAMs, cells able to contribute to the angiogenesis and fibrosis, working in concert with CAFs; (ii) HSCs activation and differentiation in myofibroblast-like cells worsening thus the fibrogenesis. Moreover, activated HSCs contribute to the immuno-tolerance, disrupting NKs functions (particularly, NKs capability to induce HSCs apoptosis) and favoring T-regs activation. These last events are facilitated also by the abovementioned TAMs able to promote a Th2 immune response, thus leading to an immuno-tolerance condition. Altogether, the above-illustrated mechanisms contribute to the HCC worsening and progression.

Moreover, ROS production represents a crucial point in the process of macrophagic differentiation into M2 and TAMs, and in fact, antioxidant treatment, reducing superoxide O2- production, inhibits TAMs differentiation and tumorigenesis in mouse models of cancer [180,181].

Moreover, TAMs can further differentiate in immunotolerant M2 macrophages, whose presence in HCC is associated with poor survival rate after curative resection [182]. These cells, in fact, through the secretion of cytokines (such as TGF-β and IL-10), chemokines, growth factors, and matrix metalloproteases, can contribute to fibrosis, tumor growth and progression, intrahepatic metastasis development, angiogenesis, and, most notably, immune suppression by promoting Th2-type immune response [182,183].

The gene expression profile of HCC patients reveals a shift from a Th1 to a Th2 immune response, suggesting a central role played by cellular immune dysregulation in hepatic tumor appearance [184]. The mechanisms underlying this change are not completely clarified yet, although different mediators produced by either stromal cells and tumour cells, such as IL-10, glucocorticoid hormones, ROS, apoptotic cells, and immune complex contribute to create a particular microenvironment that facilitates the polarization of T cells in Th2 lymphocytes [185]. This shift entails an increase of Th2 cytokines levels (i.e., IL-4, IL-5, and IL-10) and a reduction of Th1 ones (i.e., IFN-γ, IL-8, IL-15, IL-18, and IL-2). This is relevant as CD4+ Th1 lymphocytes, through production IFN-γ, enhance anti-tumoral response in HCC and accordingly, a significant decrease of CD4+ Th1-cells in patients with liver cirrhosis and HCC has been reported, highlighting their importance in liver cancer biology [186,187].

Interestingly, M2 differentiated TAMs promote immune tolerance also favoring the T-regs recruitment [178]. Tregs, characterized by the CD25 expression on their surface and by the activity of intracellular transcriptional factor forkhead box P3 (Foxp3), physiologically ensure self-tolerance by suppressing self-reactive immune cells. In particular, HCC is associated with the presence of a particular T-regs subset, namely induced T-regs (iTregs), both in the peripheral blood and in tumoral microenvironment. The iTregs depletion seems to be able to determine anti-tumoral immune responses [188,189] In this regard, the production of TGF-β and the induction of T-regs seems to be favored by myeloid-derived suppressor cells (MDSCs), a subset of inflammatory monocytes highly represented in tumoral tissue of HCC patients [190].

Moreover, an aberrant cooperation between TAMs and liver mesenchymal cells seems to be involved in the generation of a vicious cycle triggering the tumor onset. Cancer-associated fibroblasts (CAFs) can produce epidermal growth factor (EGF), hepatocyte growth factor (HGF), fibroblast growth factor (FGF), IL-6, IL-8, chemokine C-X-C motif ligand 12 (CXCL12), and matrix metalloproteases (MMP-3 and MMP-9) [179,191,192]. Through cyclooxygenase 2 (COX-2) and secreted protein acidic rich in cysteine (SPARC), CAFs recruit and stimulate TAMs, which can further increase the activation of fibroblasts through the secretion of TNF-α and Platelet-Derived Growth Factor (PDGF), generating a biologic mechanism by which phlogosis and fibrosis stimulate each other with the involvement of HSCs [193] (Figure 2). Once again, interestingly, the oxidative stress plays a central role in the biologic dynamics surrounding this process as proved by various clinical and animal models that suggest the ROS and lipoperoxidation derived products (particularly H_2_O_2_ and MDA-adducts) ability to promote the HSCs activation [194].

The activation of HSCs represents a pivotal moment in this picture. Indeed, activated HSCs can differentiate in a senescent profile, inducing the recruitment and the activation of immune cells from the bloodstream, including TAMs, as well as increasing the growth factor receptor signaling, in particular PDGF and TGF-β [195]. In line whit this, recent findings suggest the TGF-β ability to inhibit the activation and functions of Natural Killer (NK) cells by repressing the mammalian target of rapamycin (mTOR) pathway [196]. NK cells normally mediate their functions producing cytolytic granules containing perforin, granzymes, tumor necrosis factor-related apoptosis-inducing ligand (TRAIL), and IFN-γ, therefore playing a double role: cytolytic action and regulation of the other immune cells [197].

In NK cells the receptor activity, modulated by activated HSCs, inhibits the normal functioning of hepatic NK cells contributing thus to the development and progression of the liver cancer [198]. Even more in the HCC context, the NKs deficit of function may play a critical role, since these cells are able to induce apoptosis of activated HSCs, resulting by the end to exert an anti-fibrotic effect [199]. Moreover, persistent inflammation and oxidative stress generates apoptosis-resistant activated HSCs, enhancing fibrosis deposition [200].

Activated HSCs are also implicated in the promotion of immunotolerance mechanisms, secreting cytokines, which induce MDSC expansion and T-regs recruitment in tumor tissue [201]. As already mentioned, T regs in turn are able to interfere with the innate immunity, regulating NK cells and DCs activity (primarily through anti-inflammatory cytokines production such as TGF-β and IL-10) [202]. Moreover, T-regs could also disrupt the adaptive immunity response, by suppressing T-cell proliferation, CD8+ T-cells infiltration in cancer tissue, and IFN-γ secretion [203,204]. As known, IFN-γ secretion represents a fundamental moment to promote the activation of naive CD4+ and CD8+ T cells against pathogens and cancer [205]. For effective T-cell activation, in fact, the antigen presentation by Antigen Presenting Cells (APCs) to the CD4/CD8+ T-cell receptor is not sufficient [206] and the interplay between lymphocytes costimulatory receptors (g. e. CD28, ICOS, CD137, OC40, and CD27) and their relative ligands (g. e. CD80/86, B7RP1, CD137L, OC40L, and CD70) on APCs results essential [207]. Meanwhile, the physiological modulation of T-lymphocyte response is mediated by co-inhibitory receptors activated by other ligands [207]. These interactions, including CTLA-4-CD80/86 and programmed death 1 (PD-1)-PD-L1, represent the so-called “immune checkpoints”. Relevantly, chronic inflammation in NASH creates a tissue microenvironment that favors exhausted T-cells development: these cells express high levels of coinhibitory receptors (e.g. TLA-4 and PD-1) and low effector cytokines, with consequent impaired anticancer cytotoxicity [208].

Therefore, given this evidence, the possibility to regulate the immune system, nowadays is considered a very fascinating therapeutic frontier. In particular, the administration of immune checkpoint inhibitors (ICI), as anti-CTLA-4 and PD1 drugs, represents an encouraging therapeutic strategy in patients with this neoplasm [209].

Overall, these studies show how the immune system dysregulation in HCC, in terms of quantity and function alterations of immune cells, contribute to creating a tumoral microenvironment in which cancer development and progression are out of control.

In this heterogeneous scenario embracing very different cell populations, the oxidative imbalance could represent the fil rouge that connect all the elements of this inflammatory network.

## 4. Future Perspectives: The Role of Trained Immunity

By the years, the NAFLD has become one of the most important health problems in the era of obesity pandemic and strength scientific efforts have been oriented in understanding the mechanisms underlying its appearance and evolution, in which the interpretation seems to be represented by inflammation persistence, largely linked to the macrophages activation [210]. As previously mentioned, these cells could be basically involved in driving this complicated pathological picture in developing several metabolic consequences like IR, obesity, type II diabetes mellitus and atherosclerosis, all processes ground on an impaired control of the oxidative stress imbalance [211].

The medicine progress and the new advanced scientific knowledge have changed the old definition of immune system functioning. Multiple contacts with some antigens (e.g., PAMPs, DAMPs), as well as metabolic stimuli, would be able to produce a secondary immune response not exclusively made by adaptive immune cells [212,213]. According to the international scientific community, the term “Trained Immunity” (TI) represents a totality of immunological processes, which involve cells of non-specific immunity, in particular monocytes and NKs, deriving from a second antigenic contact [212]. This second contact would be able to produce the activation of several signaling pathways, which have different effects on the cellular genic expression that in turn, are mainly oriented to produce a more powerful immune reaction against antigens in comparison to the first contact [212,214].

Differently from immune adaptation which occurs in B lymphocyte, through the production of bone marrow cell clusters able to elaborate a specific response against a given antigen, this type of adaptation is mainly based on epigenetic changes, linked to a different histone’s acetylation or to the action of some siRNA like miR-155 and, therefore, it has a duration that can vary from few days to some years [212,215,216]. A TI response is not “antigen-specific” and can be transmitted to new cell generation in medullar maturation, giving them the possibility to react against the antigens with a typical “trained response” [217,218,219].

However, to better clarify the reason of a possible involvement of TI in the complex NAFLD scenario, one of the most important aspect is the interconnection between TI response and cell metabolism [220,221]. The immunometabolism represents the cornerstone research field in understanding the regulation of the inflammatory response (such as in M1 macrophages), which is supported mainly by the glycolytic activity and with a consequent stop of the Krebs cycle. Some metabolic products of this cycle such as succinate or alpha ketoglutarate play important roles as regulating mediators in this process [222].

Moreover, some enzymes of mitochondrial ETC are involved in the TI, including succinate dehydrogenase (SDH), that would represent one of the most important biological targets of itaconate [223]. Altogether, these scientific evidences confirmed the crucial role of the mitochondria in this complex biologic network by which the pathogenesis of some immune-metabolic-mediated diseases, not completely known yet, could be better clarified.

The balance between succinate and alpha ketoglutarate determines the regulation of some effectors of TI including: jumonji domain-containing (JMJ) and ten eleven translocation (TET), which would mainly act through the modification of the histones acetylation, like H3K4 [224]. To this purpose, the investigation of the monocyte mitochondrial metabolism and the cellular switch from the ETC to glycolytic prevalent energetic pathway in animal (high fat diet model) and clinical models of NAFLD could be the useful routes to fully understand the involvement of TI in this complex pathologic picture.

Moreover, another important point that could be assessed in this context is represented by the Itaconate activity, one the principal regulating mediator of TI, deriving directly from the conversion of cis-aconitate made by immune responsive gene 1 (IRG1) enzyme [225,226].

IRG-1 expression is genetically and epigenetically regulated by LPS, that encourages its expression through a pathway that involves protein kinase C (PKC) activation [227]. The miR93 whereas, decreases IRG-1 expression, by acting on the transcription factor Interferon Regulatory Factor-9 [228].

Itaconate act as an anti-inflammatory mediator, ready to regulate the unjustified and excessive activation of inflammatory cascade, acting through different mechanisms: stops the activity of SDH, involved in the Coenzyme Q synthesis and then inflammasome activation and fructose-6-phosphate kinase; forms complexes with glutathione; inhibits the transcription IkB-zeta factor, involved in the synthesis of IL-6; activates the Nrf2 through the kelch-like ECH associated protein 1 (KEAP1) alkylation [229,230]. LPS is able to induce an increase of proinflammatory cytokines production, soon followed by a stimulus for itaconate synthesis through the IRG-1 activation. In a paper recently published, it has been highlighted the importance of the TI in a mouse model of ischemia/reperfusion liver damage, used as well-known model of liver oxidative stress damage, in which the Nrf2 activity seems to be essential in this setting [231,232]. Due to the main role of macrophages in NAFLD and their essential involvement in the regulation of oxidative stress imbalance, not only in the liver but also in peripheral tissues like adipose and vascular ones, it seems to be reasonable to explore the possible involvement of the TI in this context. The assessment of Itaconate synthesis, IRG-1 expression as well as the synthesis and the biologic response derived from the activity of the above-mentioned transcription factors and/or the investigation of the effect derived from the administration, in animal models, of the itaconate, represent some important research points that are waiting to be addressed.

Moreover, in the light of the increased intestinal permeability to several immune-triggering antigens in NAFLD patients, this hypothesis acquires even more interest. This could give the opportunity not only to understand better the pathogenetic process surrounding NAFLD, but could drive the scientific community to the comprehension of its natural history and complications onset, as well as to evaluate it as a potential therapeutic target in the actual burden of the optimal medical management [233,234]

## 5. Conclusions

The complexity of NAFLD pathogenesis does not permit to individualize mechanisms incontrovertibly responsible of disease development and progression. Consequently, specific and efficient treatments are not available yet, making NAFLD a health care burden worldwide. However, a determinant contribute to the knowledge is deriving from studies on redox biology and hepatic immunity. It is well known that redox imbalance is involved in chronic liver diseases and all NAFLD stages, highlighting redox alterations as the triggering force of inflammatory response, although it is still not clear why this happens only to a part of patients. The potential involvement of pre-existing immunity factors should be explored to explain whether redox imbalance affects a susceptible immune system.

Moreover, through studies on redox biology in the immune cells, very recent reports are shedding light on the importance of immune cell metabolism, developing new concepts like trained immunity and indicating new potential therapeutic targets. Therefore, a deeper understanding of molecular mechanism underlying the crosstalk between redox biology and immunity are eagerly needed to provide a significant contribute to novel therapeutic approaches.

## Figures and Tables

**Figure 1 ijms-22-00436-f001:**
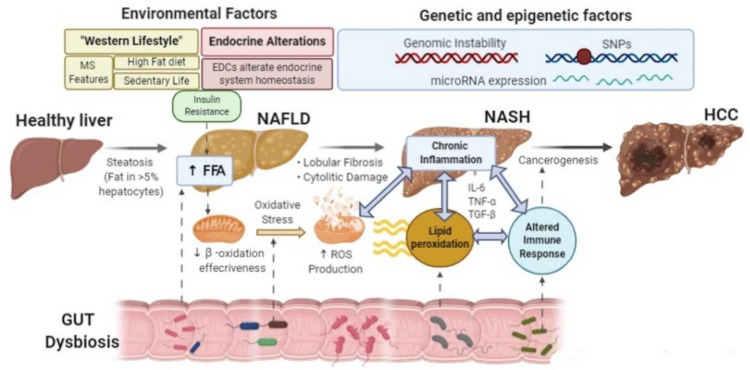
The complexity of NAFLD pathogenesis as the result of multifactorial disease.SNPs: single nucleotide polymorphisms; miRNA: microRNA expression; MS: metabolic syndrome; FFAs: free fatty acids; ROS: reactive oxygen species; EDCs: endocrine-disrupting compounds; IL-6: Interleukin-6; TNF-α: rumor necrosis factor-α; TGF-β: transforming growth factor-β; NAFLD: non-alcoholic fatty liver disease; NASH: non-alcoholic steatohepatitis; HCC: hepatocellular carcinoma. Several factors (genetic, epigenetic, and environmental) are involved in the genesis and progression of NAFLD. The onset of an insulin resistance (IR) status is a critical event contributing to the beginning as well as the worsening of the disease. In this complex scenario, oxidative stress, inflammation, impaired immune response, and intestinal dysbiosis appear altogether to be able to work in concert and promote progression, impacting the chances of an HCC occurrence.

**Figure 2 ijms-22-00436-f002:**
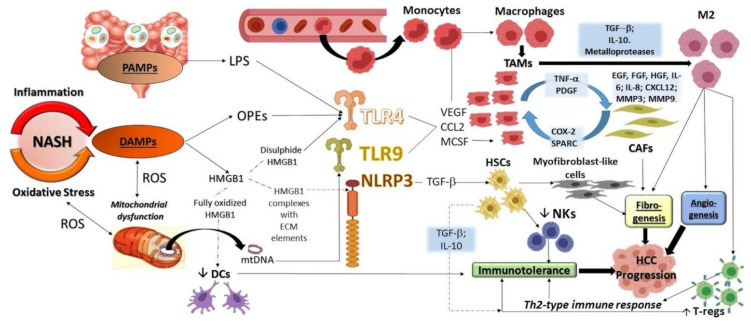
Oxidative stress, inflammation, and immune dysregulation in HCC progression. NASH: non-alcoholic steatohepatitis; ROS: reactive oxygen species; DAMPs: damage-associated molecular patterns; PAMPs: pathogen-associated molecular patterns; LPS: lipopolysaccharide; OPEs: oxidation-specific epitopes; HMGB1: High Mobility Group Box 1; ECM: extracellular matrix; NLRP3: NOD-like receptor family pyrin domain containing-3; TLR: Toll-like receptor; DCs: dendritic cells; NKs: natural killer cells; HSCs: hepatic stellate cells; CAFs: cancer-associated fibroblasts; MCSF: macrophage colony-stimulating factor; CCL2: chemokine (C-C motif) ligand 2; VEGF: vascular endothelial growth factor; EGF: Epidermal growth factor; HGF: hepatocyte growth factor; FGF: fibroblast growth factor; IL-6: interleukin 6; IL-8: Interleukin 8 chemokine; CXCL12: C-X-C motif ligand 12; MMP3: matrix metalloproteases 3; MMP-9: matrix metalloproteases 9; TGF-β: transforming growth factor-β; IL-10: Interleukin 10. Solid lines indicate direct and stable mechanisms; dashed lines indicate indirect and potential ones.

**Table 1 ijms-22-00436-t001:** Immune system-oxidative stress crosstalk main mechanisms in NAFLD progression.

Main Mechanisms and Activities	Main Effects and Epiphenomena
**Redox unbalance**: ROS ^11^ production overcomes the scavenging capacity so that superoxide anion radicals (O_2_·^−^) and hydrogen peroxide (H_2_O_2_) are continuously formed. As a consequence:	Perpetuation of chronic inflammation Creation of a vicious circle where the chronic inflammation and the oxidative stress fuelling each other, involving adaptive immune cell mechanisms, contribute to the disease progression. Increased insulin resistance Chronic inflammation worsens insulin resistance, FFAs ^1^ accumulation and thus NAFLD evolution.
Activation of several transcriptional factors and receptors (NF-κB ^6^, AP-1 ^7^, p53, HIF-1α ^8^, ^9^ PPAR-γ, β-catenin/Wnt, and ^10^ Nrf2) in immune adaptive cells determining pro-inflammatory cytokine release.	
ROS ^11^ may promote Treg ^12^ cells apoptosis, hence facilitating Th1 and Th17 pro-inflammatory activity	
KCs ^2^ and immune adaptive cells activated by ROS ^11^ (and other stimuli) become, in turn, producers of ROS ^11^.	
**OSEs ^3^:** oxidative stress promotes the production of oxidised phospholipids and aldehydes generated by lipid peroxidation such as MAA ^4^ and 4-HNE ^5^-protein adducts. As a consequence:	Inflammation and Liver Damage Pro-inflammatory cytokine release and cell death promote the worsening of liver histological damage
Induction of NF-kB ^6^ pathway promoted by MAA and 4-HNE-protein adducts in immune adaptive cells.	
**PAMPs ^13^ and DAMPs ^14^ hepatic invasion:**^13^ PAMPs (e.g., LPS ^15^), ^14^ DAMPs (e.g., HMGB1, mtDNA, lipotoxic products, and other metabolites) continuously invade the liver in NAFLD ^16^. As a consequence:	Inflammation and Liver Damage Pro-inflammatory cytokine release promotes the worsening of liver histological damage. Fibrosis By HSCs fibrogenic response.
PAMPs ^13^ and DAMPs ^14^ activate PRRs in KCs ^2^, neutrophils, HSCs ^17^, and many other cell types. PRRs ^22^ mostly involved in NAFLD are represented by: ○ TLRs ^18^: – TLR4 is activated, among others DAMPs, by HMGB1; – TLR9 is activated, among others DAMPs, by mtDNA. ○ NLRs ^19^: – NLRP3: activated, among other DAMPS, by ROS ○ OLRs ^20^: activated by nucleic acids ○ RAGEs ^21^: activated by AGEs	
TLRs ^18^, NLRs ^19^, OLRs ^20^, and RAGEs ^21^ activation leads to:: ○ Inflammosomes formation and pro-inflammatory cascades induction (e.g., MAPKs; NFkB). ○ Potential fibrogenic response (KCs ^2^ and HSCs ^17^) with collagen deposition	

^1^ FFAs: Free Fatty Acids; KCs: ^2^ Kupffer cells; ^3^ OSEs: oxidation-specific epitopes; ^4^ MAA: malondialdehyde-acetaldehyde; ^5^ HNE: dydroxinonenal; ^6^ NF-κB: nuclear factor kappa-light-chain-enhancer of activated B cells; ^7^ AP-1: activator protein-1; ^8^ HIF-1α: hypoxia-inducible factor 1-alpha; ^9^ PPAR: peroxisome proliferator-activated receptor; ^10^ Nrf2: nuclear factor erythroid 2–related factor 2; ^11^ ROS: reactive oxygen species; ^12^ Tregs: regulatory T cells; ^13^ PAMPs: Pathogen Associated Molecular Patterns; ^14^ DAMPs: damage-associated molecular patterns; ^15^ LPS: lipopolysaccharides; NAFLD: ^16^ non-alcoholic fatty liver disease; ^17^ HSCs: hepatic stellate cells; ^18^ TLRs: toll-like receptors; ^19^ NLRS: nucleotide-binding oligomerization domain-like receptors; ^20^ OLRs: oligoadenylate synthase (OAS)-like receptors; ^21^ RAGEs: receptors for advanced glycation end products; ^22^ PRRs: pattern recognition receptors.

## Data Availability

Not applicable.

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
