# Peer review of "Immunity as Cornerstone of Non-Alcoholic Fatty Liver Disease: The Contribution of Oxidative Stress in the Disease Progression"

_ijms, 2021, doi:10.3390/ijms22010436_

Round 1

Reviewer 1 Report

The review hihglighted importance of immune cell metabolism in NAFLD and describes the potential application of trained immune discoveries in disease progression. Overall, this manuscript is well written and provide with a logical flow of  molecular mechanisms that in involved in disease progression. I have some minor comments.

  1. The paragraphs of abbreviations in some pages impair the integrity. For example lines 64-68 in page the 2 and lines 312-320 in the pages 7-8. I assume that they belong the figures but it is not clear. Maybe font size could be changed and attached to the figures.
  2. Figures 1 and 2 should be improved. Some of the font size are very small and not readable.
  3. In addition to figures, maybe there could be a table summarizing the factors involved in immunity that affacts disease progression with their activities e.t.c. 
  4. There is one symbole (@-like; i.e. line 219 page 5) that used in entire manuscript but it is not quite clear what it reflects. Overall text should be checked for this symbole and should be clarified.
  5. To me, there are too many references. Some of references that are very old might be eliminated.

Author Response

Response to reviewer 1

We thank the reviewer very much for the careful revision to our manuscript. We have incorporated all his/her the suggestions that certainly improved the quality of our manuscript. In this resubmitted version, we took advantage of the criticisms received and modified the text accordingly, adding explanations and modifying the confusing section. Moreover, we submitted the paper to the revision of an English mother tongue writer to improve the readability. Following her comment: “The syntax and the lexicon are appropriate for a written scientific production, there are no grammar mistakes in this last version of the manuscript and the shortest sentences were modified or connected with other ones. It is possible to conclude that the reading paper is clear and the writing style is accurate and don’t show ambiguity”.

  1. We are completely agreed with the reviewer. It was an adaptation mistake.
  2. We modified, according with the reviewer suggestion, the figures.
  3. In accordance with the reviewer suggestion, we added a summarizing table.
  4. We corrected the mistake in all the text.
  5. In accordance with the reviewer suggestion, we modified the references section.

Reviewer 2 Report

The proposed review manuscript by Dallio et al. focuses on the problem of oxidative stress and inflammation in the context of non-alcoholic fatty liver diseases and non-alcoholic steatohepatitis. The manuscript addresses a relevant and interesting topic. However, I have several criticisms.

  1. Overall, the manuscript describes primarily reviews of other authors (approximately 80% of cited literature is review manuscripts, not original articles) and it is very difficult to understand what was shown experimentally and what the problems in the field are. Instead, I would recommend to focus on the original studies to clarify what is the gap of knowledge in the field.
  2. It is very difficult to understand what exactly the authors want to bring to a reader’s attention. Given only 4 section in the manuscript, the text is presented there as a monobloc with a mix of very different information at once (for example, page 5, paragraph 3 and 4, lines 192-209). My recommendation is to divide it in sub-blocks, for instance, “Role of mitochondria-derived ROS production in NAFLD”, “Role of ER stress in NAFLD”, “Toll-like receptors in NAFLD”, etc. I would also recommend to separate information related to NAFLD and NASH, unless the mechanisms involved in the disease’s development/progression are the same. For each block, it would be also very informative to add a separate paragraph on what is known in patients with NAFLD/NASH, including trials.
  3. Please improve quality of the figures. In the present view it is very difficult to read text in the figures. I would suggest using very user-friendly software provided by BioRender (biorender.com). Also, please add description for each figure aiming to help a reader to understand what is shown on a figure. Please correct position of the abbreviations key for your figures, as currently they are located somewhere in the text (page 2, lines 64-68 and page 7-8, lines 312-321).
  4. When you say, “Several further discrepancies exist in the literature about the mitochondrial dysfunction in simple steatosis and NASH such as…” (page 4, lines 129-133) a reader expects to get information on what those discrepancies are. Instead, after you have finished all the enumerations, you continue accidently with ER stress. Please correct it and describe what is known in the literature and what the problem is. And this is, actually, a problem of the entire manuscript: switching attention from one molecular to another, from one condition to another, from one mechanism to another without a logic reasoning.
  5. Page 6, line 247: please add reference to the study you cite here.
  6. Block 4 “Future perspectives: the role of Trained Immunity”: instead of showing the future perspectives in the field, this block continues with simple description of what was possible to find in the literature about immunity with a weak connection to the topic of the manuscript at the end (page 10). Please correct it.
  7. Please add to the “Conclusion” what problems and unresolved questions remain in the field.

Author Response

Response to reviewer 2

We thank the reviewer very much for the careful revision to our manuscript. We have incorporated all his/her the suggestions that certainly improved the quality of our manuscript. In this resubmitted version, we took advantage of the criticisms received and modified the text accordingly, adding explanations and modifying the confusing section.

  1. In accordance with the reviewer suggestion, we modified the references section reducing the redundancies and focusing the attention on the recent published original articles properly referred to the specific fields in each sentence.
  2. We thank the reviewer for these suggestions which improved the quality of reading. Accordingly, we have divided paragraphs 1.1 and 2 in two sub-paragraphs in order to better orient the reading through the text (1.1.1 Oxidative stress derived from mitochondrial disfunction; 1.1.2 ER contribution to oxidative stress; 2.1 The role of PRRs; 2.2. The role of OSEs). We do not believe that a further subdivision in NAFLD and NASH would benefit the manuscript, as NASH is the evolutive condition of NAFLD and not a distinctive disease. In accordance, the aim of the manuscript is to elucidate the mechanisms involved in NAFLD progression. Regarding the last point, accordingly with Editorial board Guidelines we focused only on molecular mechanisms as this is adherent to the journal scope, hence avoiding additional information about clinical trials. However, human data to support molecular theories were shown along the lines.
  3. We modified, according with the reviewer suggestion, the quality of the figures; in particular, we used the software “BioRender” to improve font size text, cells and receptors representation as well as pathogenesis events illustration. We repositioned the description and the abbreviations of each figure appropriately.
  4. We thank the reviewer for the comment. We specified the content of studies mentioned in the sentence. (Lines 138-141).
  5. We thank the reviewer for her/his suggestion. However, if the reviewer was referring to this sentence “We have previously described the importance of the bidirectional relationship between oxidative stress and immunity in NAFLD progression”, there is no study mentioned here.
  6. We thank the reviewer for his/her careful comment that give us the opportunity to clarify better the reason of the paragraph “future perspectives” structure. Due to the novelty of the topic, in particular if we consider it in such as a disease molecular model like NAFLD (never explored before), it is reasonable to elucidate in the first part of the paragraph the main biologic characteristics of the trained immunity. It acquires a very big relevance because represents the key of lecture for the understanding of the reader and doesn’t represent a “mere academic exercise”. We highlighted in the middle part of the paragraph (line 430-463, old version of the manuscript) the molecular mechanisms surrounding this process that could be potentially involved properly in NAFLD context, giving to the reader a scientifically sound prospective view of what is could be better to explore in future experiments. However, following the reviewer suggestion we clarified better this aspect in order to improve the overall quality of this section as requested.
  7. We thank the Reviewer for the suggestion. We improved the “Conclusions” section

Reviewer 3 Report

This manuscript was well summarized including the new insight. It is acceptable for publication with minor modifications. The position of "GUT dysbiosis" looks only related to NASH to HCC. It is better to move the adequate position. In addition, "Genetic and Epigenetic factors" is also better to move suitable position. The possibility of ICI (immune checkpoint inhibitors) is better to include for the treatment option to HCC.

Author Response

Response to reviewer 3

We thank the reviewer very much for the careful revision to our manuscript. We have incorporated all his/her the suggestions that certainly improved the quality of our manuscript. Moreover, we modified the references section, reducing the redundancies and focusing the attention on the recent published original articles.

  1. We are completely agreed with the reviewer. In order to highlight its potential role and implication in whole pathogenesis, we modified the figure 1 moving “GUT dysbiosis” in a more central position. Besides, we repositioned “Genetic and Epigenetic Factors” in a more suitable position too.
  2. We included the possibility of ICI as treatment option for HCC at the end of the specific section.

Round 2

Reviewer 2 Report

In the revised version of the manuscript the authors addressed all the concerns raised from the first submission, and the manuscript is ready for publication as it is. I just would like to pay attention of the authors that they need to acknowledge BioRender in the text, according to the requirements from their website to be able to publish a graphical image created with BioRender software.